# Shannon Entropy as an Indicator for Sorting Processes in Hydrothermal Systems

**DOI:** 10.3390/e22060656

**Published:** 2020-06-13

**Authors:** Frank J. A. van Ruitenbeek, Jasper Goseling, Wim H. Bakker, Kim A. A. Hein

**Affiliations:** 1Department of Earth Systems Analysis, Faculty of Geo-Information Science and Earth Observation (ITC), University of Twente, Hengelosestraat 99, 7500 AA Enschede, The Netherlands; w.h.bakker@utwente.nl; 2Faculty of Electrical Engineering, Mathematics and Computer Science (EEMCS), University of Twente, Drienerlolaan 5, 7522 NB Enschede, The Netherlands; j.goseling@utwente.nl; 3School of Geosciences, University of the Witwatersrand, Private Bag 3, Johannesburg 2050 Wits, South Africa; kim.hein@wits.ac.za

**Keywords:** Shannon entropy, hydrothermal, processes, sorting, chemistry, minerals, early life

## Abstract

Hydrothermal processes modify the chemical and mineralogical composition of rock. We studied and quantified the effects of hydrothermal processes on the composition of volcanic rocks by a novel application of the Shannon entropy, which is a measure of uncertainty and commonly applied in information theory. We show here that the Shannon entropies calculated on major elemental chemical composition data and short-wave infrared (SWIR) reflectance spectra of hydrothermally altered rocks are lower than unaltered rocks with a comparable primary composition. The lowering of the Shannon entropy indicates chemical and spectral sorting during hydrothermal alteration of rocks. The hydrothermal processes described in this study present a natural mechanism for transforming energy from heat to increased order in rock. The increased order is manifest as the increased sorting of chemical elements and SWIR absorption features of the rock, and can be measured and quantified by the Shannon entropy. The results are useful for the study of hydrothermal mineral deposits, early life environments and the effects of hydrothermal processes on rocks.

## 1. Introduction

Hydrothermal processes affect the chemical and mineralogical composition of rock by destabilizing and breaking down the primary rock mineralogy and by forming new secondary minerals [1,2]. The types of newly formed alteration assemblages and their chemical compositions are determined by the hydrothermal fluid chemistry and physical parameters such as temperature and pressure. The hydrothermal processes cause depletion and enrichment of selected elements that change the relative proportions of chemical elements and minerals in the altered rock [3] and result in sorting of elements and minerals in rocks. 

Enrichment and depletion of elements and minerals play an important role in the formation of hydrothermal ore deposits. Enrichment of elements and minerals may lead to the formation of economic accumulations of metals and minerals [4]. Changes in the elemental and mineralogical composition of rock by the interaction of hydrothermal fluids also occur in association with environments that are considered to be favorable for the creating and sustaining early life forms [5,6].

The effects of hydrothermal processes on the composition of rocks are commonly studied by measuring the absolute or relative concentrations of chemical elements [7,8] and by identifying and quantifying mineral assemblages, in particular, those of the newly formed alteration minerals, e.g., [9]. The resulting descriptions and measurements of both the chemical and mineralogical composition provide information on the type and intensity of the hydrothermal alteration processes [10]. None of the commonly used methods, however, provide single quantitative measures of the degree of depletion and enrichment of chemical elements, or the breakdown and formation of alteration minerals. For objective comparisons between observations within and between areas, and in order to quantify the effect of hydrothermal processes on composition we require single quantitative measures.

In this study, we use a novel application of the Shannon entropy to study and measure the effects of hydrothermal processes on the composition of the rock. The Shannon entropy was originally developed in the context of digital transmission of information [11,12]. In information theory, the Shannon entropy is used as a measure of uncertainty in probability density distributions. In this study, Shannon entropy is used to measure the effects of hydrothermal processes on rock by comparing the Shannon entropies calculated on measurements of hydrothermally altered rocks with those of their unaltered precursors. We present and discuss the Shannon entropies that were calculated from major element compositional data and the short-wave infrared (SWIR) reflectance spectra of the rock samples, the chemical Shannon entropy (H_CHEM_) and the spectral Shannon entropy (H_SPEC_), respectively. We will show that the Shannon entropy provides single quantitative measures of the effects of hydrothermal processes on the composition of rocks, and we will explain that the lowering of the Shannon entropy in the hydrothermally altered rocks is the result of sorting processes.

## 2. Materials and Methods

Three different rock sample sets were used in this study. One set represents the rocks that were altered by hydrothermal processes. The composition of the unaltered precursors could not be measured directly from these rocks because of the intense alteration. Therefore, two sample sets were used as analogs of the unaltered precursors to estimate the chemical and spectral mineralogical composition. The three sample sets are described below. The method of calculating the Shannon entropy of these rocks sets is also explained.

### 2.1. Rock Sample Sets

The hydrothermally altered rocks are represented by a suite of 10 samples from volcanic lithologies of the Soansville greenstone belt of the East Pilbara Granite-Greenstone (EPGG) terrane in Western Australia [13]. The samples were collected from the hydrothermally altered footwall of the Kangaroo Caves Volcanogenic Massive sulfide mineralization [14]. The samples are chlorite-quartz and quartz-sericite altered volcanic rocks of originally dacite-rhyodacitic and andesite-basaltic composition. The Soansville greenstone belt was intensely altered by hydrothermal processes at around 3.24 Ga, i.e., during the MesoArchean [15]. The rock samples were described by petrographic methods. SWIR reflectance spectra were obtained from cut rock surfaces using an ASD FieldSpec Pro spectrometer in the range from 350 to 2500 nm. Whole-rock chemical compositions were measured by a lithium metaborate/tetraborate fusion ICP-OES after crushing, splitting and pulverizing using mild steel. The chemical analyses were performed by Activation Laboratory Inc. Eight international reference materials were included in the analysis for data quality purposes.

The chemical composition of precursor analogs is represented by the analyses of a suite of 176 volcanic rocks from several Archean greenstone belts of the EPGG terrane in Western Australia [16]. The selection of 176 samples consists of greenschist metamorphic volcanic rocks of basaltic to rhyolitic composition, and with ages ranging from approximately 3.51 to 3.31 Ga [17]. Care was taken to only select samples whose chemical composition was not affected by alteration processes (Appendix A). Due to the relative similarity in age, and the absence of chemical modification, these rock samples are considered to be useful analogs to the precursors of the hydrothermally altered rocks mentioned above. The chemical composition of the rocks was analyzed by X-ray Fluorescence spectrometry on fused disks. The trace element Zr was measured on a pressed pellet [16]. 

The SWIR reflectance spectra of analogs of the unaltered precursor rocks are represented by spectra of a suite of 61 unaltered volcanics from the ASTER spectral library [18], ranging in composition from basaltic to rhyolitic. The spectra were measured by a Perkin-Elmer Lambda 900 UV/VIS/NIR Spectrophotometer from 0.4 to 2.5 μm on fine powders. ASTER spectral library samples were chosen since no spectrally unaltered volcanic precursors of the hydrothermally altered rocks were available from the EPGG terrain in Western Australia.

### 2.2. Shannon Entropy

The Shannon entropy (*H*) of a probability distribution (*P*) is defined as [11,12]:
(1)$$H\left(P\right)=-\overset{m}{\underset{i=1}{\sum }}p_{i}\times log_{2}p_{i}$$
where *m* is the number of possible outcomes and *p_i_* is the probability of outcome *i*. The units of Shannon entropy are bits, because of the log_2_ term in the equation.

The Shannon entropy is used as a measure of uncertainty in probability distributions. Maximum uncertainty, or maximum Shannon entropy, occurs in a distribution where all possible outcomes have equal probabilities. Such a distribution resembles maximum heterogeneity or randomness. Minimum uncertainty is reached when one of the possible outcomes has a probability of one. In that case, there is no uncertainty and the Shannon entropy becomes zero. Such a situation can be regarded as a form of minimum randomness. Shannon entropies were calculated on distributions of the major element chemical compositions, H_CHEM_, and on SWIR reflectance spectra, H_SPEC_, of hydrothermally altered rocks and their assumed precursors. 

#### 2.2.1. Chemical Shannon Entropy (H_CHEM_)

Shannon entropies of the elemental chemical composition distribution (H_CHEM_) were calculated from the whole-rock chemical compositions of the hydrothermally altered and metamorphic volcanic rocks from the EPGG terrane. This type of entropy was named the chemical Shannon entropy (H_CHEM_). The following 10 major elements, expressed in weight percentages of their oxides, were included in the calculations and normalized to 100%: Si, Ti, Al, Fe, Mn, Mg, Ca, Na, K and P. Less than detection values were replaced by half of the detection limit values following common practice. The element concentrations were first converted to molar percentages of the oxides and subsequently into molar percentages of the single elements. The chemical composition of each sample was then converted to a probability distribution by rescaling of the concentrations from a sum of 100% to 1. The H_CHEM_ of the composition of each sample was calculated using Equation (1). Trace and volatile elements were not included in the calculation. 

The precise chemical composition of the actual precursors of the hydrothermally altered rocks cannot readily be determined because of the high degree of alteration that had changed the primary mineralogy. Therefore, the precursor composition was inferred using the Zr/TiO_2_ ratio, which is a good indicator of the precursor composition of volcanic rocks even when the rock is intensely altered [19]. A comparison of the Zr/TiO_2_ ratios of the hydrothermally altered rocks to the Zr/TiO_2_ ratios (and related H_CHEM_ values) of the metamorphic volcanic rocks from the EPGG, was used to infer H_CHEM_ of the actual precursor of the altered rock. The relationship between the H_CHEM_ and the Zr/TiO_2_ ratio of the metamorphic volcanic rocks was calculated by linear regression of the means of the Zr/TiO_2_ ratios (predictor) on the H_CHEM_ (predicted) of the different volcanic lithologies. Since the number of rock samples across the lithological groups is highly unbalanced, the means of each lithological group were used for this regression instead of the samples themselves. The resulting model was applied to estimate the H_CHEM_ from the Zr/TiO_2_ ratios of the hydrothermally altered volcanic rocks (Appendix B).

#### 2.2.2. Spectral Shannon Entropy (H_SPEC_)

Shannon entropies were calculated from the SWIR spectra of the hydrothermally altered rocks and the unaltered rocks from the ASTER spectral library. This type of entropy was named the spectral Shannon entropy (H_SPEC_). The reflectance spectra of the two sample sets were first resampled to a 1 nm spectral sampling and equal wavelength ranges. The resulting spectra covered the 1300–2500 nm range in 1201 discrete bands. This wavelength range contains diagnostic vibrational absorption features of many hydrothermal alteration minerals [20]. The reflectance spectra were converted to absorption by subtraction of all reflectance values from 1, because the entropy was measured on absorption spectra. The resulting absorption spectra were normalized to 1 and subsequently used to calculate H_SPEC_ following Equation (1) for each rock sample.

Similar to the H_CHEM_, the spectral entropy H_SPEC_ of the precursor of the hydrothermally altered rock could not readily be determined from the altered rock. Therefore, the H_SPEC_ was inferred from representative rock spectra of the ASTER spectral library of the different types of unaltered rocks. The relationship between spectra and volcanic rock composition is more straightforward than in the case of the chemical composition since the different precursor rocks have almost identical spectra. Therefore, a statistical approach to assess H_SPEC_ of the inferred precursors was not applied. 

#### 2.2.3. Change in Shannon Entropy

The difference between the entropy of the hydrothermally altered rock and its precursor was calculated for each rock by:
(2)$$\Delta H=H_{Hydrothermally\;altered\;rock}-H_{Unaltered\;precursor\;rock}$$


The difference ΔH was calculated for both H_CHEM_ and H_SPEC_, resulting in ΔH_CHEM_ and ΔH_SPEC_.

## 3. Results

### 3.1. Chemical Shannon Entropy

The Shannon entropies calculated from major element chemical data (H_CHEM_) of hydrothermally altered rocks are lower than those of the unaltered precursor rocks (Figure 1). Within the groups of altered rocks there is a difference between the two types of alteration facies; quartz-sericite altered rocks have lower H_CHEM_ values than chlorite-quartz altered rocks. The difference in entropy is largely related to higher contents of Si, a slight increase in K in the hydrothermally altered rocks, and lower contents of Ca and Na (see Figure 2a,b for the mean compositions of the two groups of altered rock). Also, the quartz-sericite altered rocks contain low contents of Fe, Mn and Mg compared to their precursor rocks. The mean H_CHEM_ values of the groups of unaltered precursor rocks decrease with a more felsic composition, which is caused by an increase in Si, Na and K and a decrease in Ti, Fe, Mn, Mg and Ca (Figure 2c–f). The decreasing H_CHEM_ of the unaltered rocks (Figure 1) is the result of progressive magma fractionation from basaltic to rhyolitic rock. All calculated H_CHEM_ values of the two rock sample sets are significantly lower than the maximum Shannon entropy of 3.32 bits of a distribution of 10 possible outcomes with equal probabilities. A one-way analysis of variance (ANOVA) test confirmed that the mean H_CHEM_ values of the four lithological and two alteration groups are not similar (*f*-value = 498.7, *p*-value < 0.001). A pairwise comparison using the Tukey’s range test showed that the differences between the six group are statistically significant for all pairs (*p*-value < 0.001) except for the rhyolite and chlorite-quartz pair (*p*-value = 0.052).

### 3.2. Spectral Shannon Entropy 

The Shannon entropies calculated on SWIR reflectance spectra (H_SPEC_) of the hydrothermally altered rocks are lower than those of the unaltered precursor rocks (Figure 1). The quartz-sericite altered rocks have significantly lower H_SPEC_ values than the unaltered rock, while the H_SPEC_ values of the chlorite-quartz altered rocks are slightly lower compared to the unaltered rocks. The H_SPEC_ values of the unaltered rocks are close to the maximum Shannon entropy of 10.23 bits of distributions with 1201 spectral samples. The high H_SPEC_ values indicate that the reflectance spectra of these rocks approximate distributions of near-equal probabilities. This can be observed in Figure 3 where reflectance spectra of the unaltered rocks are devoid of absorption features and have an almost horizontal line. An exception is the spectrum of rhyolite_H1 that contains a shallow absorption feature around 1900 nm due to water molecules in the rock. The presence of the absorption features produces a decrease of the Shannon entropy. 

The SWIR spectra of quartz-sericite altered rocks show relatively high reflectance values and deep absorption features. This is caused by the abundances of sericite, which produces deep absorption features and high reflectance values of the spectral hull (Figure 3). The absorption features are produced by O-H bonds near 1400 nm, bonds in H_2_O and/or O-H near 1900 nm, and Al-OH bonds near 2200, 2350 and 2440 nm. The variation of H_SPEC_ values within the group of quartz-sericite altered rocks is due to differences in absorption feature depths and the overall values of the spectra. The SWIR spectra of the chlorite-quartz altered rocks are dominated by chlorite. Chlorite causes absorption features by O-H bonds near 1400 nm, bonds in H_2_O and/or O-H near 1900 and 2000 nm, and Fe-OH and Mg-OH bonds near 2350 nm and 2450 nm. The spectra also show absorption by Al-OH bonds near 2200 nm caused by minor sericite. The shallower absorption features and the overall lower values of the spectra in chlorite-quartz altered rocks result in higher H_SPEC_ values compared to those of the quartz-sericite altered rocks and slightly lower values than those of the unaltered rocks. The H_SPEC_ values decrease by increasing depth of absorption features, higher reflectance values of the hull, and deviations from horizontal and flat spectral shapes. The mean H_SPEC_ values of the groups of unaltered precursor rocks do not vary systematically and are not significantly different.

### 3.3. Changes in the Shannon Entropies

The differences in chemical and spectral Shannon entropy between the hydrothermally altered rocks and the unaltered precursor rocks (ΔH_CHEM_ and ΔH_SPEC_, Equation (2)) are shown in Figure 4. For an explanation of the estimation of the chemical Shannon entropy of the unaltered precursors of the altered rocks see Appendix B. The quartz-sericite and chlorite-quartz altered rocks cluster in different parts of Figure 4, which means that the degree of sorting differs between the two groups of altered rocks. The quartz-sericite altered rocks show the highest degree of chemical and spectral sorting, which is obvious from the relatively high ΔH_CHEM_ and ΔH_SPEC_ values that range between 0.969 and 1.232 bits and between 0.034 and 0.074 bits, respectively. The chlorite-quartz altered rocks show less elevated values of ΔH_CHEM_ and ΔH_SPEC_ values, which range between 0.363 and 0.612 bits and between 0.001 to 0.002 bits, respectively. The decrease in Shannon entropy of the hydrothermally altered rocks represents a lowering of the uncertainty in the probability distributions of the major element composition and the SWIR reflectance spectra from which the Shannon entropy was calculated. We interpret this decrease in uncertainty as a form of information that was measured using the distributions of rock measurements, and imposed on the rock itself, by the hydrothermal processes and which can be measured using the Shannon entropy.

## 4. Discussion

### 4.1. Interpretation of the Shannon Entropy

The chemical Shannon entropy H_CHEM_ represents the uncertainty in the type of chemical element measured when one atom of the rock is sampled. The uncertainty with respect to the type of selected atom is larger when the distribution has equal probabilities than in the situation when the distribution has more varying probabilities. The latter occurs in the suite of hydrothermally altered rocks where selective enrichment and depletion resulted in distributions of high probabilities of Si and low probabilities of most of the other elements. A low chemical Shannon entropy is interpreted as the result of increased sorting of chemical elements in the altered volcanic rock.

The spectral Shannon entropy H_SPEC_ is interpreted as the uncertainty in wavelength of absorbed IR radiation when one IR-ray that has been absorbed by the rock is sampled and measured. The uncertainty in the wavelengths at which absorption occurs, decreases in SWIR spectra with deep absorption features in a few narrow wavelength ranges in bright rock. In these rocks, there is a dominance of absorption at a few narrow wavelength ranges. Flat horizontal spectra approach equal probabilities and produce high entropy values, indicating the absence of deep absorption features. The low spectral Shannon entropy is interpreted as the result of increased sorting of absorption features in the altered rock. From other studies [21], it is known that deeper absorption features are caused by increased abundances, smaller grain-sizes and better surface exposure of the SWIR active minerals as is the case for the quartz-sericite altered rocks. 

The Shannon entropy provides quantitative estimates of the effects of sorting processes on the composition of rocks. It is important to note that the Shannon entropy is somewhat subjective since the values depend on the type and number of the measured variables. We do not normalize the Shannon entropy and therefore there is no maximum bound. For quantitative comparisons between rocks analyzed in different batches or from different areas, the measurement parameters have to be standardized. This means that parameters, such as the number and type of chemical elements or spectral bands and the units in which they are measured, must be equal. Different measurement parameters will give different Shannon entropy values of the same rock.

### 4.2. Relationship between Heat, Hydrothermal Alteration and Shannon Entropy

By placing our results in a wider geological context of the study area, we found a relationship between the change in Shannon entropy of the hydrothermally altered rock and the heat of a cooling magma that drove the hydrothermal system in which the rocks were altered. The altered rocks in this study were originally deposited in a submarine seafloor environment where heat provided by a coeval sub-volcanic intrusion drove hydrothermal fluids through the volcanic sequence [15]. Temperatures of up to about 450 °C occurred in the basal parts of the sequence [22]. 

Circulation of predominantly seawater-derived fluids caused large-scale alteration of the volcanic rocks, where the type and composition of the altered rock depend on the fluid composition and physicochemical conditions. The hydrothermal fluids destabilized the primary volcanic minerals, such as volcanic glass and ferromagnesian minerals, and produced quartz-sericite and chlorite-quartz as alteration minerals. Alteration reactions caused the liberation of Na and Ca, which were subsequently removed by hydrothermal fluids, and the accumulation of Si. Further depletion of Fe and Mg occurred in zones of sericite-quartz alteration. Fe and/or Mg in chlorite-quartz altered rock were retained in chlorite. The breakdown of precursor minerals and the formation of new minerals changed the chemical composition and increased the sorting of elements in the volcanic rock. 

Due to similarity between the Shannon entropy and the statistical thermodynamic entropy of Boltzmann [23,24], i.e., both measure the number of possible configurations of components in a system, the Shannon entropy may potentially be used as an indication for parts of the thermodynamic entropies of hydrothermal environments. However, it has to be kept in mind that the two types of entropy are conceptually different. The Shannon entropy is based on the uncertainty in probability distributions, while the thermodynamic entropy is a state definition of a physical system. Exploring relationships between the Shannon and thermodynamic entropies in hydrothermal systems is a direction for further study. Shannon entropies can also be employed to quantify the degree of order in the shape and spatial arrangement of minerals and aggregates in rock by calculations on microstructural parameters [25]. Calculating such spatial entropies is also a direction for further research. 

### 4.3. Mineralized and Early Life Environments

Sorting processes play an important role in the formation of mineral deposits, where selective enrichment and depletion may lead to the accumulation of elements or minerals. The Shannon entropy is a measure of the degree of sorting of chemical elements in rock and can, therefore, be used to detect these accumulations. The Shannon entropy is insensitive to the types of elements that are enriched and depleted. The method is complementary to conventional methods of rock composition analysis and does not replace them. Many mineralized environments are formed by hydrothermal processes, where hydrothermally altered rocks are associated with economic accumulations of elements or minerals. The Shannon entropy can act as a proxy for mineralization by enabling the identification of zones of intense wall-rock alteration, independent of the type of alteration.

Hydrothermal environments are considered favorable for developing and sustaining early life [5,26]. Hydrothermal alteration in these environments produces rocks of low spectral Shannon entropies. These rocks may influence the radiative environment by absorption of radiation at specific wavelengths and by providing uniformity in the vibration frequencies of molecular bonds within crystal lattices in the rock [27]. Since the radiative environment plays an important role for early life in harvesting energy for metabolic processes, there may be a relationship between the low Shannon entropy of rocks and the maintenance of low entropic states that are typical for living organisms [28].

## 5. Conclusions

We conclude that the hydrothermal processes described in this study present a natural mechanism for transforming energy from heat to increased order in rock. The relationship between heat and Shannon entropy is indirect and based on changes in the probability distributions of rock measurements. The increased order is manifest as increased sorting of chemical elements and SWIR absorption features of the rock and can be measured and quantified by the Shannon entropy. The results are useful for the study of hydrothermal mineral deposits, early life environments and the effects of hydrothermal processes on rock.

## Figures and Tables

**Figure 1 entropy-22-00656-f001:**
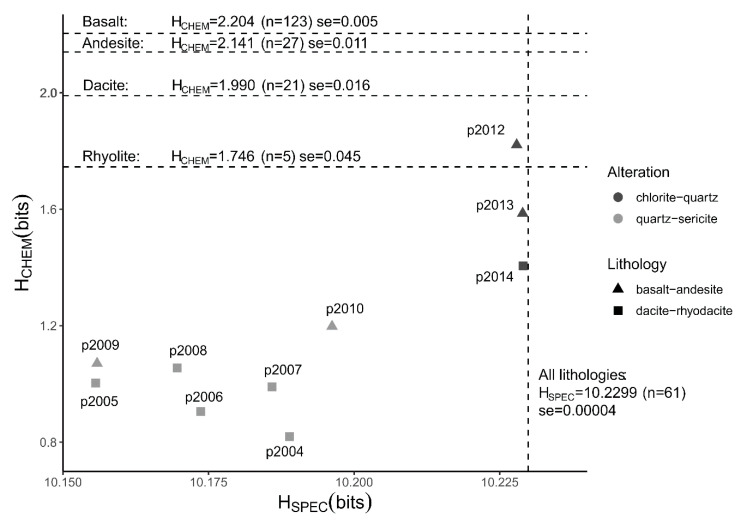
Spectral and chemical Shannon entropy of hydrothermally altered and unaltered rocks. Plot shows Shannon entropies calculated from short-wave infrared (SWIR) spectra (H_SPEC_) and from chemical compositions (H_CHEM_) of hydrothermally altered rocks. The type of alteration is shown in greyscale. The type of volcanic rock is shown using different symbols. Labels show sample numbers of the hydrothermally altered rocks. Dashed lines show mean entropies of unaltered precursor rocks of different composition. Mean entropy, number of samples (*n*) and the standard error of the mean (se = standard deviation/($\sqrt{n}$) are shown for each group of precursor rocks. Chemical and spectral data of unaltered rocks were taken from Smithies, Champion, van Kranendonk and Hickman [16] and Baldridge, Hook, Grove and Rivera [18], respectively.

**Figure 2 entropy-22-00656-f002:**
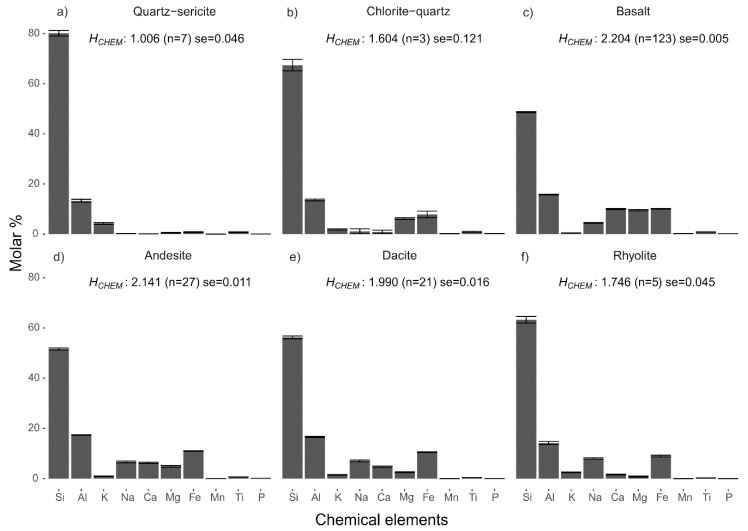
Chemical composition of altered and unaltered rocks. Histograms of mean chemical compositions of hydrothermally altered rocks (**a**,**b**) and unaltered precursors (**c**–**f**). Mean entropy, number of samples (*n*) and the standard error (se = standard deviation/$\sqrt{n}$) are shown for each group of rocks. Error bars show the mean + standard error of the mean element concentrations. Chemical data of unaltered rocks were taken from Smithies, Champion, van Kranendonk and Hickman [16].

**Figure 3 entropy-22-00656-f003:**
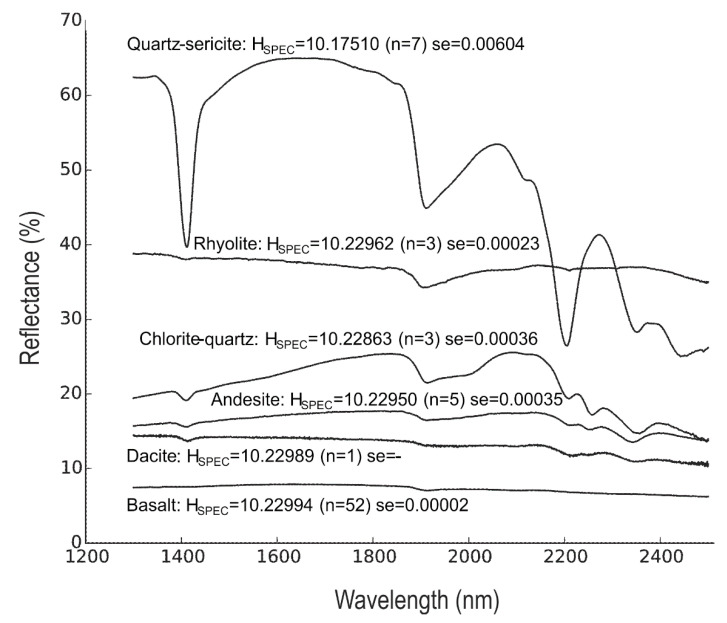
SWIR reflectance spectra of altered and unaltered rocks. Mean reflectance spectra of hydrothermally altered rocks (quartz-sericite and chlorite-quartz) and unaltered precursors (basalt, andesite, dacite and rhyolite). Mean entropy, number of samples (*n*) and the standard error) are (se = standard deviation) shown for each group of rocks. Spectra of unaltered rocks were taken from Baldridge, Hook, Grove and Rivera [18].

**Figure 4 entropy-22-00656-f004:**
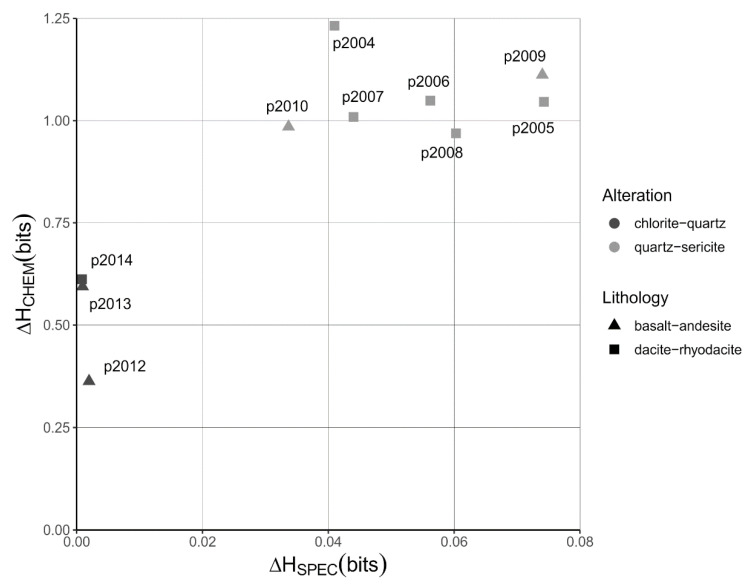
Quantification of the sorting effects. Plot of differences in Shannon entropies calculated from SWIR reflectance spectra (ΔH_SPEC_) and chemical compositions (ΔH_CHEM_) between hydrothermally altered rocks and their precursors. These differences in Shannon entropy represent the measurements of the sorting effects on the altered rocks. The type of rock alteration is shown in greyscale. The type of volcanic rock is shown using different symbols. Unaltered rocks plot near the origin because their compositions have not changed. Labels show sample numbers of the hydrothermally altered rocks.

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
