# Peer review of "Shannon Entropy as an Indicator for Sorting Processes in Hydrothermal Systems"

_entropy, 2020, doi:10.3390/e22060656_

Round 1
Reviewer 1 Report
This is an interesting manuscript where an entropic approach is taken to characterise differences in geologically modified rocks. I am not a geoscientist so I cannot judge how much -- if any -- impact this work has in that field, but it's an interesting read to a non-expert, too. It's also a nice and original idea.
The paper is well-written but a serious flaw: it is not explained how the Shannon entropy H is actually calculated. It is rigorously bound between [0,1] when p_i's are properly normalised. In the manuscript, however, there are values of H larger than unity. This must be because of (improper) normalisation? The authors need to explain what exactly the p_i's are that are used in the calculations and how the normalisation is done.
Reviewer 2 Report
The paper compares the difference in chemical composition of various rocks due to hydrothermal process. Samples are collected and their chemical composition is described by a) the fractions of 10 chosen elements or b) their spectral composition in a discretization of the spectrum in 1201 band windows. In both cases the authors deal with a normalized histogram or discrete probability distribution p. The Shannon entropy H(p) associated to the discrete distribution is computed and it is found that in the general case it is lower for rocks that have undergone the hydrothermal process.
This represent an interesting piece of information and the approach presented in the papers is (according to the authors) new, so the paper is interesting.
However, in my opinion the paper could be improved if some misleading interpretations of Shannon entropy contained in the paper are clarified.
For example the authors speak of “entropy of the rocks” (line 55) or of “information imposed to the rock” (line 253).
They need to avoid confusion between the Shannon entropy of the sample H(p) and the physical entropy of a thermodynamic system, for example S(e,v). They are different concepts. H(p) is useful for comparing the information content of a probability distribution, but it contains a degree of subjectivity due to the choice of the experimenter.
For example the authors find that the entropy of the samples after the hydrothermal process is lower because there is a increase in the Si content and hence the distribution is more “concentrated” on a single element. Had they decided to exclude the Si and consider only the fractional composition of the remaining 9 elements, probably the could have observed (at least in some cases) an increase of entropy as a result of the hydrothermal process (compare histograms a) and c) of Fig. 2). This should caution on extending the statements on the sample distribution to the entire population or physical system. So I suggest to expand and clarify the discussion around lines 166, 167 accordingly.
The same ambiguity is contained in the Conclusions (lines 304) where the authors link the change in Shannon entropy of the samples with the heath transfer undergone by the rock. The relation between heat supply and entropy change concerns the physical entropy S(e,v) and not the Shannon entropy H(p). For example, suppose that the samples are opinion polls before the elections between 10 political parties. An external event may influence the polls in order to polarize the choices of the electors towards a single party and hence lower the entropy of the samples. But no one could infer that this is due to the heat extracted form the population.
Another point that should be clarified is the relation between the collected samples and the population. Can we say that the different samples are extracted by the same population or that there are different populations? If there are different samples of the same population, how is measured the internal variability between the samples? Measuring only the standard deviation of their entropy may be misleading because samples with different chemical compositions may have the same Shannon entropy. Simple statistical tests like chi-squared test could be used to check if different samples are extracted from the same population.
Minor corrections.
line 280, 281. the statement “ the Shannon entropy may potentially used as a proxy for parts of thermodynamical entropy’s…” is unclear and should be re-thinked. Also I suggest to avoid the use of the expression “proxy of” because is slightly colloquial (see line 294).
line 107-110. The use of “uniformity” proposed is the opposite of the common use for describing a probability distribution with nearly equal probabilities. I suggest to avoid this use of “uniformity” in the paper because is cumbersome and it is used only once or twice in the paper.
Final remark: In the paper various detailed explanations are given to explain the difference in chemical composition of the rock due to the hydrothermal process. These are rather technical and difficult to follow for the readers of Entropy Journal and they are of a completely different nature with respect to the Information entropy approach introduced in the paper. I suggest to clearly separate in the paper with different sections the geological analysis from the information theory considerations.
Round 2
Reviewer 1 Report
The manuscript can now be accepted for publication.
Reviewer 2 Report
The authors addressed my concerns and now the manuscript is clearer